# Consortium between Groundwater Quality and Lint Yield in Cotton Belt Areas

Muhammad Rashid Hameed [1], Houneida Attia [2], Umair Riaz [3], Kamran Ashraf [4], Khalid H. Alamer [5], Ashwaq T. Althobaiti [2], Badreyah Algethami [2], Khawar Sultan [1], Aamir Amanat Ali Khan [1] and Qamar uz Zaman [1,*]

1   Department of Environmental Sciences, The University of Lahore, Lahore 54590, Pakistan
2   Department of Biology, College of Science, Taif University, P.O. Box 11099, Taif 21944, Saudi Arabia
3   Department of Soil and Environmental Sciences, MNS-University of Agriculture, Multan 66000, Pakistan
4   Department of Food Sciences, Government College University Faisalabad, Sahiwal Campus, Sahiwal 57000, Pakistan
5   Biological Sciences Department, Faculty of Science and Arts, King Abdulaziz University, Rabigh 21911, Saudi Arabia
*   Correspondence: qamar.zaman1@envs.uol.edu.pk

**Abstract:** The agriculture sector of Pakistan mainly depends on freshwater from groundwater resources. Deterioration of these resources adversely affected crop yields due to climate change and human activities. A comprehensive study was conducted to evaluate the groundwater quality of varying boring depths and the possible effects on the crop yield of cotton in Tehsil Fort Abbas, District Bahawalnagar, Punjab, Pakistan. A total of 347 samples were collected from the investigated areas. Results revealed that 86% of samples were declared unfit for irrigation purposes, 6% of samples were fit, and 8% of samples were marginally fit for irrigation. The ranges for the electrical conductivity (EC), sodium absorption ratio (SAR), and residual sodium carbonate (RSC) were 0.61–10.49 dS m$^{-1}$, 0.65 to 5.44 meq L$^{-1}$, and 0.02 to 5.44 meq L$^{-1}$, respectively. Regarding the EC of water samples, the southwestern side of the study area where the lower values were observed was in an acceptable range in terms of water quality. Differential response to metal contamination was observed in the study area. Lower contamination of metals was observed in the water samples collected from some regions on the eastern and western sides of the study area. Principal component analysis (PCA) showed that by increasing the depth of the bore, the value of EC was also increased. Similarly, for the cotton lint yield maximum yield (1040 kg acre$^{-1}$) was observed in the sampling point located in the southwestern part of the study area due to better quality of irrigation water, while the minimum cotton lint yield (520 kg acre$^{-1}$) was noticed in sampling point located in the western side of the study area. Overall groundwater quality of Tehsil Fort Abbas was unfit for irrigation due to the high EC values and metal concentrations. The yield showed a negative correlation among all parameters of water. It was suggested that using the recommended dose of gypsum powder/stone and dilution of groundwater with canal water reduced the hazards of anions and cations of groundwater for the accumulation of salts in crops.

**Keywords:** water quality; metal accumulation; boring depth; sodium absorption ratio; cotton lint yield

## 1. Introduction

Water is compulsory for the survival and maintenance of the human population and economic development [1]. The agriculture sector of Pakistan dominantly depends on groundwater, and where there is no access to river water, people use well water for drinking and irrigation purposes [2]. Water is used for various purposes, including growing crops and making products in factories. In Pakistan, which is a farming country, about 66% of the total population depends on farming [3,4]. Groundwater is playing an essential role in expanding irrigated agriculture in many parts of the world. Pakistan is the third-largest

user of groundwater for irrigation in the world [5]. The surface water supplies are sufficient to irrigate 27% of the area, whereas the remaining 73% is directly or indirectly irrigated using groundwater. In the Punjab province, more than 90% of the total groundwater is abstracted [6]. Currently, 1.2 million private tube wells are working in the country, out of which 85% are in Punjab, 6.4% are in Sindh, 3.8% are in Khyber-Pakhtunkhwa, and 4.8% are in the Baluchistan Province [4]. The total groundwater extraction in Pakistan is about 60 billion cubic meters [7]. It is common knowledge that all irrigation water contains dissolved minerals and salts; however, the concentration and composition of these salts may vary depending on the source of the water, the depth of boring, and the season [8]. The farmer must be aware of the concentration and composition of irrigation water at various times of the year since salts can harm plant growth. The chemical weathering of minerals is the primary cause of salts in irrigation water (from rocks and soils) [9]. Over millions of years, much of the salt in geological formations has been naturally dissolved and carried by water. Aquifer containing fresh water also dissolves salts from minerals [10].

About 10 billion cubic meters of groundwater was used in Pakistan in a year by the end of 1965, which has increased to about 68 billion cubic meters in 2002 [11]. The groundwater level in Pakistan is about 20 to 30 m deep. About 17% of the area in Punjab Province of Pakistan is irrigated with groundwater [4,12]. In Pakistan, water is a significant source, about 90% of water is used for irrigation, which provides about 45% of employment and contributes to 25% gross domestic product (GDP) of the country's requirements [13]. In Pakistan, 26% of the land consists of desert from total arid land, and a total of 21 million hectares (m ha) is cultivated land, of which about 15 m ha consists of irrigated land and the other 6 m ha under dry agriculture [14]. An ecosystem is affected by using groundwater and vice versa, according to ECGD & WFD (European Commission Groundwater Directive and Water Framework Directive) [2,15]. In agricultural areas, land cover and land use changes are basic parameters to alter groundwater level and quality [2,16]. Pollution of groundwater consists of two parts; one is the deterioration of groundwater quality and decreases in level due to overuse of groundwater, and the second is a bad impact on people due to the use of contaminated groundwater [17]. Crop residuals, agrochemicals, municipality waste, and natural sources of nitrogen are the foremost sources of increasing nitrogen content in groundwater [2,4,18].

Irrigation is necessary for crop production in arid and semi-arid areas. The irrigation water must not contain soluble salts at levels detrimental to plants or have a negative impact on the soil quality [19]. Most of the time, there is not enough water of such high quality to meet the needs of crops that are farmed [2,4]. In these circumstances, farmers are compelled to use irrigation water with high levels of dissolved salts or high residual sodium carbonate (RSC), which inevitably results in lower crop yields [2,12]. Unwise use of this water can frequently result in crop failures and the development of saline or sodic soils, which then necessitate costly remediation to restore their ability to support plant growth [20]. About 70% of ground and surface water was contaminated in Pakistan, according to a recent report by Syed et al. [21]. The quality of groundwater in several major cities of Punjab provinces, i.e., Lahore, Faisalabad, Multan, Sheikhupura, Bahawalpur, and Bahawalnagar, is badly affected due to the poor management of factories disposal and sewage drainage [22]. Due to the deterioration of groundwater quality, the soil quality is badly affected, which has resulted in the decline of the annual yield of crops [2,4,12].

The quality of irrigation water and any possible long-term effects on agricultural crops of unfit/brackish water must be given serious consideration. Two-thirds of the rural population is dependent on groundwater for their food security and livelihood, directly or indirectly. The government needs to take critical steps to protect the rights of smallholder farmers. In the management of groundwater, one of the major bottlenecks is the lack of vigorous analysis of the data to comprehensively understand the dynamics of groundwater use in agriculture and its impacts on the socio-economic conditions of the farmers and the environment. The quality of groundwater is poor in most areas of Pakistan, and its continuous use leads to soil degradation, which affects crop yield and creates a

serious problem of salinization. To this end, this experimental approach hypothesizes that the groundwater quality is affecting the yield of cotton in the Tehsil Fort Abbas, District Bahawalnagar, southern Punjab. The main objectives of this research work were to: (1) assess the groundwater quality and toxic heavy metals concentration from an irrigation viewpoint using hydro-chemical analysis; (2) investigate the impact of tube well irrigation on cotton crop yield; check the spatial distribution of groundwater quality using the GIS tool.

## 2. Materials and Methods

### 2.1. Study Area Description

The investigated area was Tehsil Fort Abbas, located in District Bahawalnagar, southern Punjab, Pakistan. The boundaries of District Bahawalnagar in east and south touched the Indian border while Bahawalpur district is situated on its west and Sutlej River flows on its northern side. The total area of District Bahawalnagar is about 8878 square kilometers, encompassing five tehsils and 118 Union Councils. Tehsil Fort Abbas is situated on the border of Pakistan and India, south of Haroonabad Tehsil, near Faqirwali. Fort Abbas tehsil is situated in Bahawalnagar district, district of South Punjab, Pakistan, located 29°11′33″ North and 72°51′13″ East geographically [23]. The Elevation of this city is 163 m (535 ft.) above sea level. The border of India is situated 5 to 6 km from the eastern side of the city of Fort Abbas. The study area is bounded by forests, deserts, and agricultural land. The population of Tehsil Fort Abbas was about 423,529 people by the census of 2017 [24]. The total land area of Tehsil Fort Abbas is about 254,113 acres, out of which forests occupied 47 acres of the total land. The cultivated land of the study area is 221,924 acres, 32,142 acres is a non-cultivated land area, and 21,150 acres of land is arid. Orchids in the study area occupied 1700 acres. Hakra is the only canal that provides most of the water to the area and is further divided into Hakra Left (H.L) and Hakra Right (H.R), which enters into the tehsil from north–east. Annual precipitation in the study is recorded to be a maximum of 203 mm (8.01 inches) and a minimum of 1.0 mm (0.04 inches), and the temperature has been recorded as a maximum of 50.1 °C and a minimum of −1.0 °C. The foremost crops in this area are cotton, wheat, and mustard [25].

### 2.2. Water Sampling

A total of 347 samples were collected of groundwater from different locations in the study area. Samples of groundwater were collected from different agricultural areas of Tehsil Fort Abbas by tube wells and turbines. While collecting samples, it has been considered that the turbine is running for a 30 min minimum. Pre-cleaned plastic bottles (polypropylene) were used to collect the water samples and stored them for laboratory analysis. The water samples were labeled according to their source, time, and region from which they were obtained. Samples were stored at a 15 °C temperature until they were transported into the research laboratory for analysis. GPS coordinates, the yield of cotton, depth of the borehole, and area irrigated by turbine were recorded at the time of sampling. The map of the sampling locations is shown in Figure 1.

### 2.3. Analysis of Samples

Collected samples were taken to the Soil and Water Testing Laboratory Bahawalpur for analysis. For the determination of the pH of water samples, a pH meter (pH 200 Senso-direct) and for the electric conductivity (EC), a conductivity meter (CON200 Senso-direct, Lovibond, Tintometer GmbH, Dortmund, Deutschland) was used by following the standard protocols. Concentrations of $Na^+$ and $K^+$ were determined using flame photometry (PFP-7, Jenway, UK). The Ca, Mg, $HCO_3$, $CO_3$, and chloride contents were measured by using the titration method (EDTA trimetric). Heavy metals (Cd, Mn, Pb, Cu, Zn, and Fe) were determined by using the atomic absorption spectrometer AAS (AI 1200). All the hydro-chemical analysis of irrigation water was performed by following the protocols devised by Richards [26] and Umair et al. [4] and instrument manuals. The sodium adsorption

ratio (SAR) and residual sodium carbonate (RSC) were calculated by using the equations given below.

$$SAR = Na^{+1} \Big/ [(Ca^{+2} + Mg^{+2})/2]^{1/2} = meq/L \qquad (1)$$

$$RSC\ (meq/L) = \left(CO_3^{-2} + HCO_3^{-1}\right) - \left(Ca^{++} + Mg^{++}\right) \qquad (2)$$

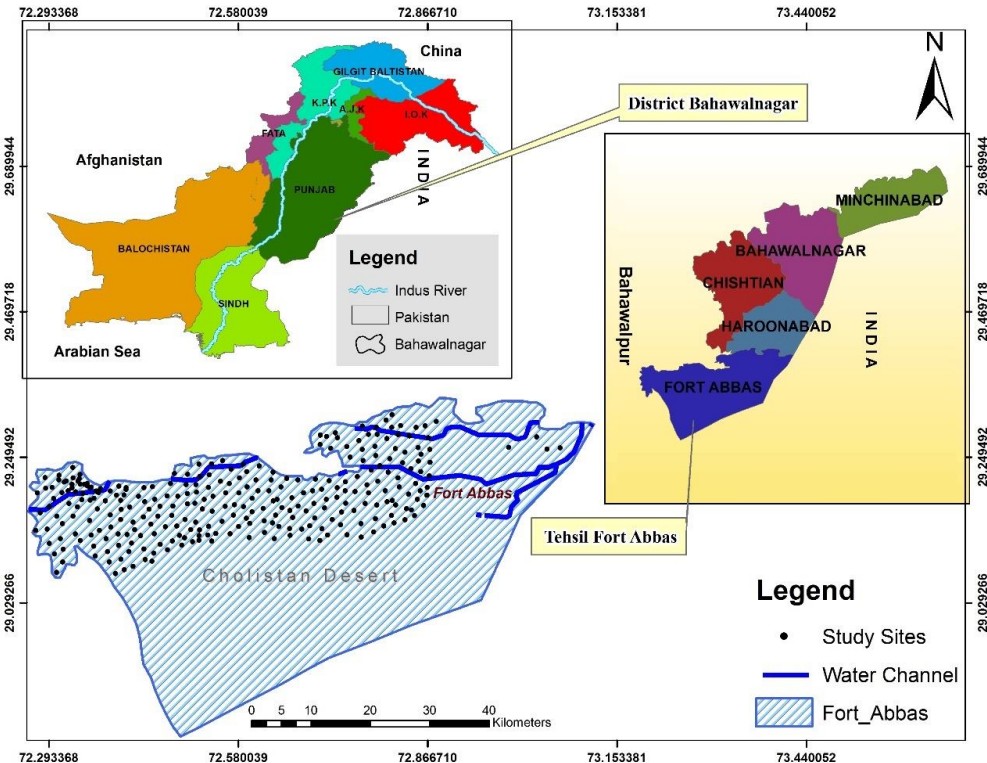

**Figure 1.** GIS locations of water sampling and cotton yield studied area.

### 2.4. Geographical Information Services (GIS)

GPS coordinates were recorded with a location accuracy of ±3.0 m. Maps of different parameters of the quality of groundwater in the study area are shown and developed using the ArcGIS 10.3 software. Maps showed results with respect to fit, marginal fit, and unfit by different colors and marks.

### 2.5. Statistical Analysis

Basic statistical analysis of the collected data and graphical representations of all attributes were performed by using Microsoft Excel 2016. All the treatment means were compared by using Statistics 8.01 [27]. Principal component analysis (PCA) was carried out by using XLSTAT software, and biplots were generated to compare the correlation among the observed data. The separate correlation analysis was performed using R software.

## 3. Results

### 3.1. Electrical Conductivity (dS/m) Status

The standard value of EC ranged from 0 to 1.45 dS/m. The EC value of the fit samples ranged between 0 and 1.15 dS/m. It was noticed that 84% of samples were unfit, 9% samples were marginal fit, and 7% samples were found to be fit based upon standards defined by the FAO for electrical conductivity criteria for irrigation water (Figure 2).

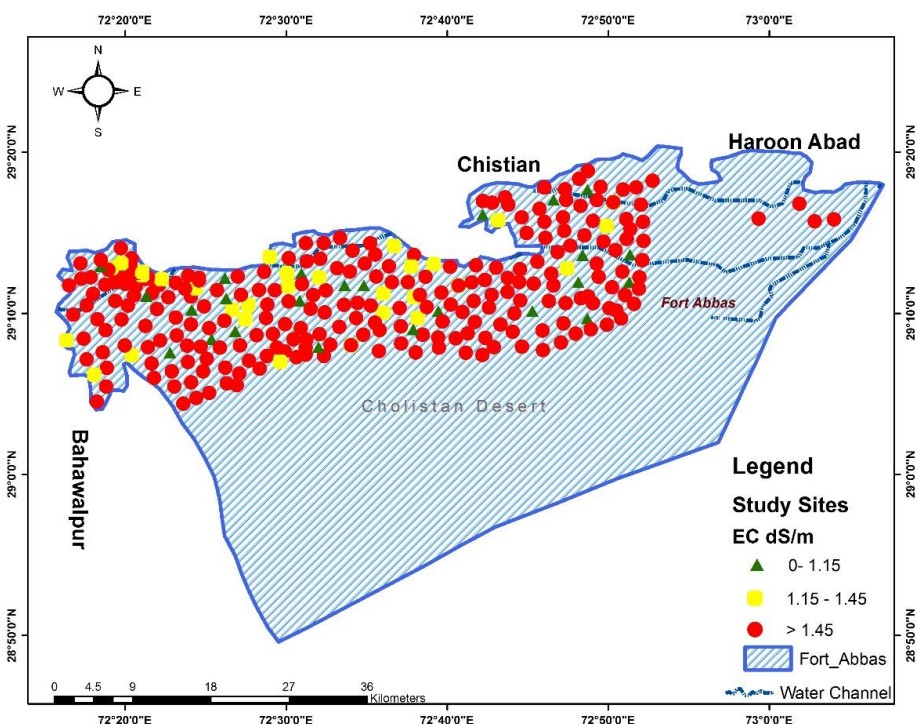

**Figure 2.** Status of electrical conductivity (EC) of groundwater in Tehsil Fort Abbas.

### 3.2. Sodium Absorption Ratio SAR (meq/L)

According to the Food and Agriculture Organization (FAO), the standard value of SAR for irrigation water ranges from <6 to >10 mmol/L. However, results showed that the observed SAR ranged from 0.65 to 24.62 mmol/L. Furthermore, it was noticed that 26% of samples were unfit, 28% samples were marginal fit, and 46% samples were fit based upon SAR criteria for irrigation water according to the standards defined by the FAO (Figure 3).

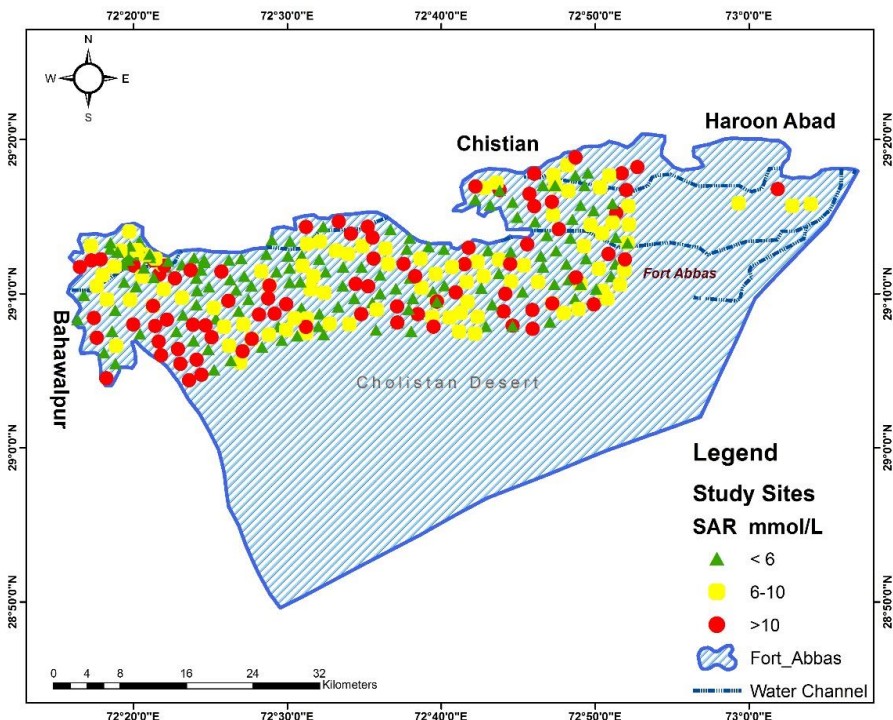

**Figure 3.** Status of sodium adsorption ratio (SAR) of groundwater in Tehsil Fort Abbas.

### 3.3. Ratio of Residual Sodium Carbonate (meq/L)

According to the Food and Agriculture Organization (FAO), the standard value of RSC for irrigation water varied from <1.25 to >2.5 meq/L. However, results showed that the observed RSC ranged from 0.02 to 5.44 meq/L (Figure 4). Moreover, it was noticed that 6% of samples were unfit, 6% samples were marginal fit, and 8% samples were fit. The RSC value of the fit samples was found to be <1.25 meq/L according to the standards defined by the FAO (Figure 4).

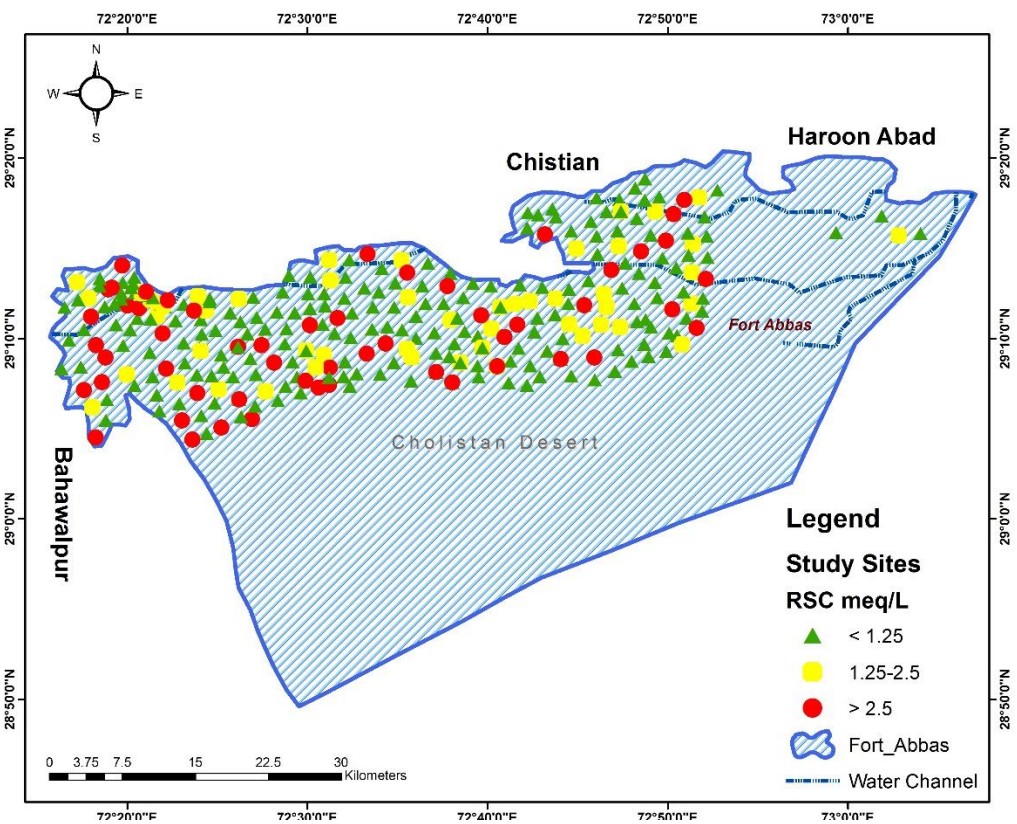

**Figure 4.** Status of residual sodium carbonate (RSC) of groundwater in Tehsil Fort Abbas.

To represent the quality status, the criteria for each studied parameter are shown in Table 1. A total of 347 samples were collected, and 299 samples were considered to be unfit for irrigation, which was 86% of the total samples. Only 21 samples were found to be fit for irrigation the purpose; its percentage was 6% of the total samples. One interesting thing noted was that 27 samples lie in the category of marginal fit and the percentage of these samples was 8% of the total samples (Figure 5).

**Table 1.** Criteria for the qualification of irrigation water for fitness.

| No. | Parameters | Fit | Marginal Fit | Unfit |
|-----|-----------|-----|--------------|-------|
| 1 | EC dS/m | 0 to 1.15 | 1.15 to 1.45 | > 1.45 |
| 2 | SAR mmol/L | < 6 | 6 to 10 | > 10 |
| 3 | RSC meq/L | < 1.25 | 1.25 to 2.5 | > 2.5 |

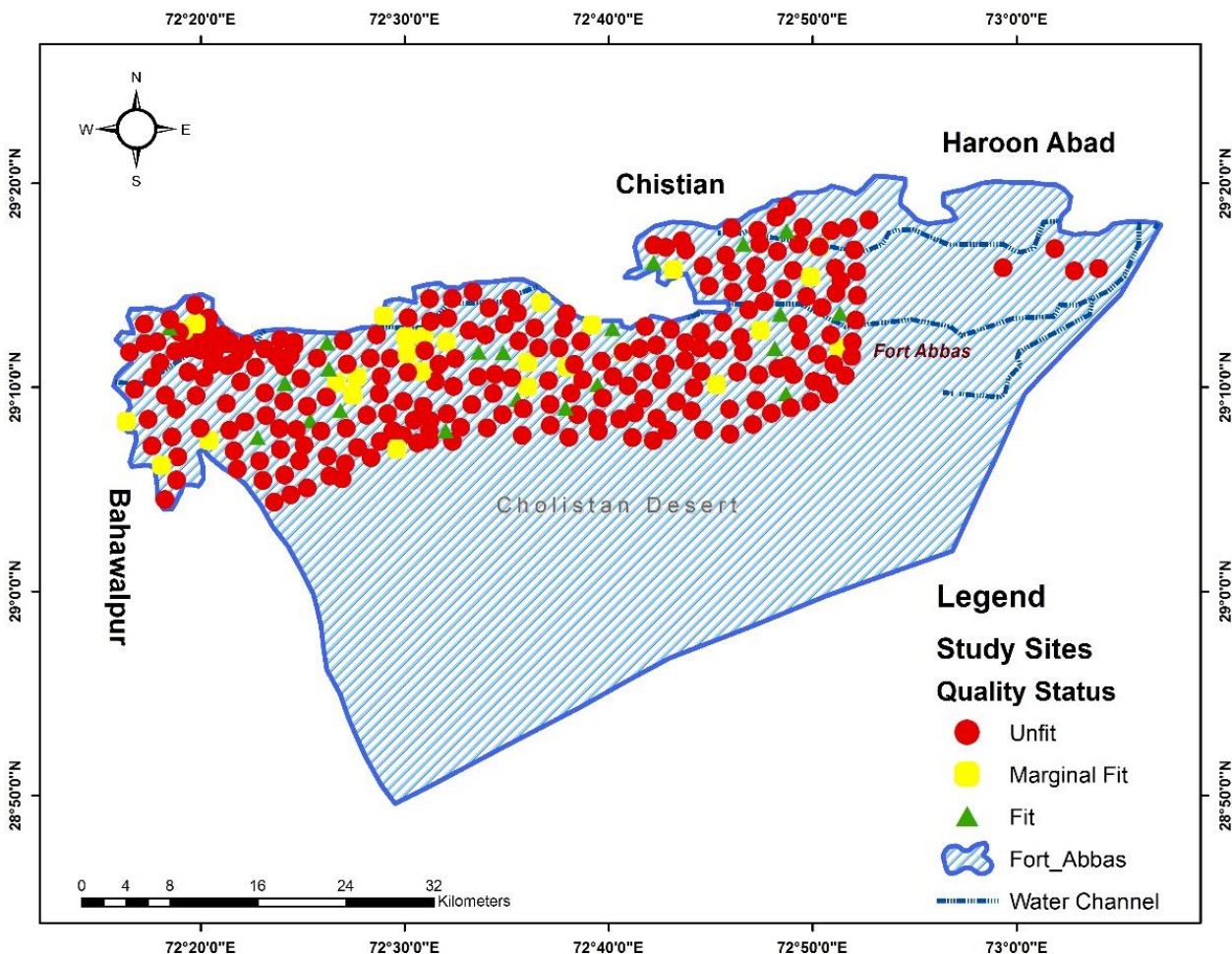

**Figure 5.** Irrigation water quality status of groundwater in Tehsil Fort Abbas.

*3.4. Metal Analysis of Groundwater*

The standard value for Cu, Zn, Fe, Mn, Cd, and Pb contents as described by the FAO for irrigation water is less than or equal to 0.20, 2.00, 5.00, 0.20, 0.01, and 0.1 ppm, respectively. The percent response for the fitness of water samples in relation to the heavy metals was represented in Figure 6. Minimum Cu content of 0.01 ppm and a maximum value of Cu content of 3.25 ppm were measured in this study; however, Zn contents ranged between 0.01 and 4.06 ppm. For Cu and Zn contents, it was noticed that 12% and 82% of samples were fit for irrigation, and 88% and 12% of samples were unfit, respectively. The Fe contents varied from 0.03 to 9.03 ppm. It was observed that 87% of samples were fit for irrigation, and 13% of samples were unfit with respect to Fe standard in the study area. The concentrations of Mn contents ranged between 0.03 and 9.75 ppm in the irrigation water samples of the study area. It was observed that 2% of samples were found to be fit for irrigation, and 98% of samples were unfit with respect to Mn standards described by the FAO. Similarly, minimum Cd and Pb content was measured to be 0.01 ppm for both, and the maximum was measured to be 3.25 and 3.36 ppm, respectively. It was observed that 0.2% and 5% of samples were fit for irrigation, and 99.7% and 95% of samples were unfit with respect to the Cd and Pb standards suggested by FAO, respectively.

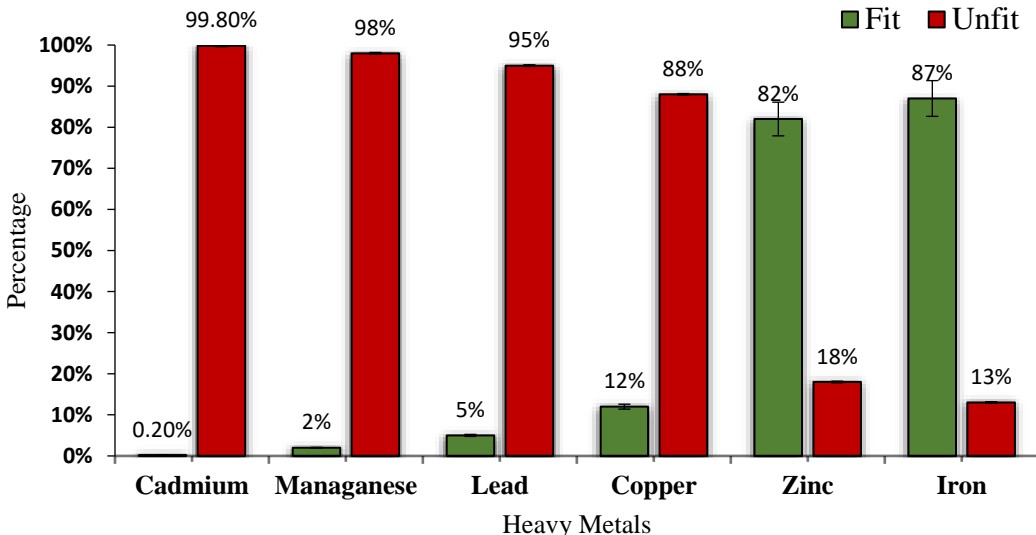

**Figure 6.** Status of heavy metals in collected water samples from Tehsil Fort Abbas.

The overall descriptive analysis of various variables of the irrigation water is depicted in Table 2. The overall trend of the variables of the current study was also compared with Ayers and Westcot [28].

**Table 2.** Descriptive analysis of studied variable of the irrigation water of sampling sites with the permissible limits.

| Variables | Range | Mean | S.D. | C.V. | Skew | 1st Quartile | 3rd Quartile | None | S to M | Never |
|---|---|---|---|---|---|---|---|---|---|---|
| | | | **Present Study** | | | | | | **Degree of Restriction on Use [28]** | |
| EC | 0.62–10.49 | 2.90 | 1.68 | 58.05 | 1.24 | 1.72 | 4.13 | <0.7 | 0.7–3.0 | >3.0 |
| Ca + Mg | 1.48–1919.0 | 19.06 | 115.89 | 607.87 | 14.61 | 7.10 | 12.06 | - | - | - |
| Bicarbonates | 4.08–17.40 | 8.25 | 2.35 | 28.54 | 0.57 | 6.42 | 9.88 | <1.5 | 1.5–8.5 | >8.5 |
| Sodium | 1.18–1957.0 | 23.57 | 104.91 | 445.07 | 18.11 | 8.56 | 24.80 | <3 | 3–9 | >9 |
| Cl | 0.12–40.0 | 3.31 | 3.08 | 93.01 | 5.64 | 1.92 | 4.12 | <4 | 4–10 | >10 |
| SAR | 0.65–483.0 | 9.03 | 25.93 | 287.19 | 17.66 | 4.32 | 10.55 | >0.7 | 0.7–0.2 | >0.2 |
| RSC | 0.00–5.44 | 0.39 | 1.02 | 263.44 | 3.05 | 0.00 | 0.00 | <1.5 | 1.50–2.50 | >2.50 |
| Cd | 0.01–3.25 | 1.03 | 0.69 | 67.43 | 0.52 | 0.36 | 3.25 | - | - | >0.01 |
| Cu | 0.01–3.25 | 1.03 | 0.70 | 67.38 | 0.52 | 0.36 | 1.45 | - | - | >0.20 |
| Fe | 0.03–9.03 | 2.88 | 1.94 | 67.40 | 0.52 | 1.00 | 4.03 | - | - | >5.0 |
| Zn | 0.01–4.06 | 1.29 | 0.87 | 67.36 | 0.52 | 0.45 | 1.81 | - | - | >2.0 |
| Pb | 0.01–3.36 | 1.07 | 0.72 | 67.39 | 0.51 | 0.37 | 1.50 | - | - | >5.0 |

EC = electrical conductivity; Cl = chloride contents; SAR = sodium absorption ratio; RSC = residual sodium carbonate; Cd = cadmium contents; Cu = copper contents; Fe = iron contents; Zn = zinc contents; Pb = lead contents; S.D. = standard deviation; C.V. = coefficient of variation; S to M; slight to moderate.

*3.5. Effect of Groundwater Quality on Annual Cotton Yield*

The average cotton yield in the studied areas is shown in Table 3. The data revealed that the average cotton yield was decreasing every year due to a shortage of river water supply and the overuse of low-quality groundwater. The maximum cotton yield was recorded to be 1040 ± 65.32 kg/acre from the point located on the southwestern side of the study area, followed by 733.34 ± 49.9 Kg/acre located in the western side of the study area, while minimum cotton yields were recorded as 493 ± 82.19 Kg/acre in the sampling site located on the eastern side of the experimental site. The main reason behind this was that most of the samples collected from the southwestern side of the experimental side were found to be fit for irrigation purposes according to standard values.

**Table 3.** Average cotton yield with respect to investigated area.

| Sampling Points | Location in Study Area | No. of Respondents | Average kg/Acre | Water Quality Status |
|---|---|---|---|---|
| Chak no. 316 h/r marot | South Western Side | 36 | 1040 ± 65.32 | Fit |
| Chak no. 329 h/r marot | Western Side | 26 | 760 ± 65.32 | Marginal Fit |
| Chak no. 325 h/r marot | Western Side | 15 | 520 ± 65.32 | Unfit |
| Chak no. 341 h/r marot | Western Side | 17 | 906.67 ± 82.19 | Marginal Fit |
| Chak no. 313 h/r marot | Eastern Side | 18 | 493.4 ± 82.19 | Unfit |
| Chak no. 310 h/r marot | Eastern Side | 12 | 920 ± 97.98 | UnFit |
| Chak no. 338 h/r marot | Southwestern Side | 12 | 866.67 ± 82.19 | Unfit |
| Chak no. 328 h/r marot | Western Side | 16 | 946.67 ± 82.19 | Marginal Fit |
| Chak no. 319 h/r marot | Western Side | 41 | 933.34 ± 67.99 | Marginal Fit |
| Chak no. 314 h/r marot | Eastern Side | 27 | 760 ± 65.32 | Unfit |
| Chak no. 317 h/r marot | Southwestern Side | 32 | 1000 ± 65.32 | Fit |
| Chak no. 326 h/r marot | Western Side | 18 | 733.34 ± 49.9 | Unfit |
| Chak no. 315 h/r marot | Eastern Side | 17 | 826.67 ± 99.8 | Unfit |
| Chak no. 340 h/r marot | Western Side | 2 | 920 ± 97.98 | Unfit |
| Chak no. 318 h/r marot | Western Side | 17 | 960 ± 97.98 | Marginal Fit |
| Chak no. 324 h/r marot | Western Side | 13 | 813.34 ± 82.19 | Unfit |
| Chak no. 204 h/b alif walhar | Eastern Side | 15 | 760 ± 65.32 | Unfit |
| Chak no. 327 h/r marot | Western Side | 3 | 813.34 ± 82.19 | Unfit |
| Chak no. 312 h/r marot | Eastern Side | 9 | 946.67 ± 82.19 | Marginal Fit |
| Chak no. 311 h/r marot | Eastern Side | 1 | 840 ± 86.41 | Unfit |

### 3.6. Principal Component and Correlation Analysis

Principal component analysis showed that EC was highly dependent on the depth of boring (Figure 7). When the depth of bore of a tube well or turbine was increased, it decreased the value of EC, chloride, and carbonate in water samples. The sodium absorption ratio was also dependent on the depth of the borehole of a turbine. PCA pointed to several clusters based on metal contents, major cations and anions (saline water), and the water extraction depth. Deeper the water extraction depth, the lower the dissolved salts and, therefore, the better the quality of irrigation water. A cluster of desert soils plots away from the main groups is marked by the low yield of crops. It is located on the eastern side of the study area, including part of the Cholistan desert (sandy soils). The high-yield areas are located away from the desert soils on the western side of the study area. A general trend of increasing crop productivity is shown by the arrows in the study area

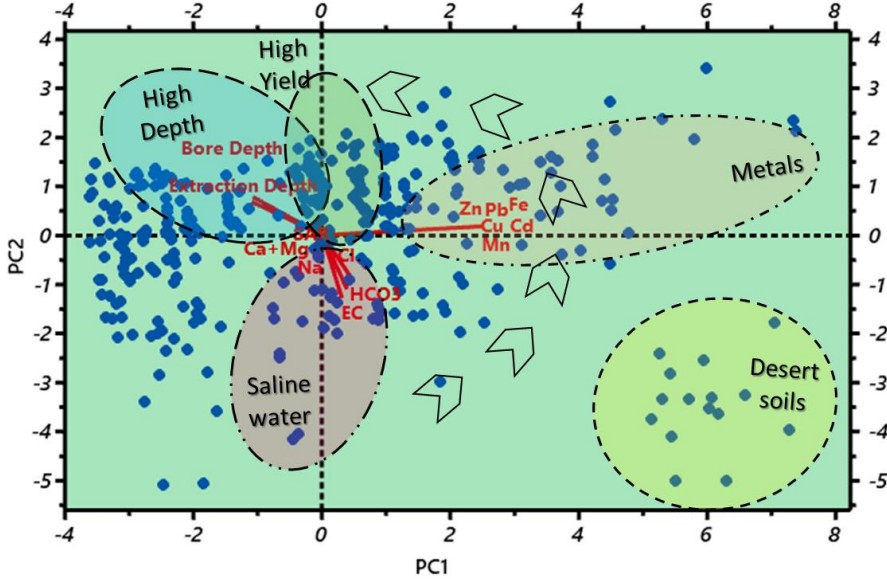

**Figure 7.** Principal component analysis between physicochemical attributes of groundwater.

Cluster two do not show any influence on cluster one or any other components. The statistical analysis explored that cotton yield showed a negative correlation among all studied parameters. The EC and $Ca^{++}$ showed a strong positive correlation (0.81), while $CO_3$ and $HCO_3$ showed a moderate positive correlation (0.63). The correlation data revealed that with the increment in EC of irrigation water, the crop yield decreased with a similar trend in the case of $Na^+$ and SAR (Figure 8).

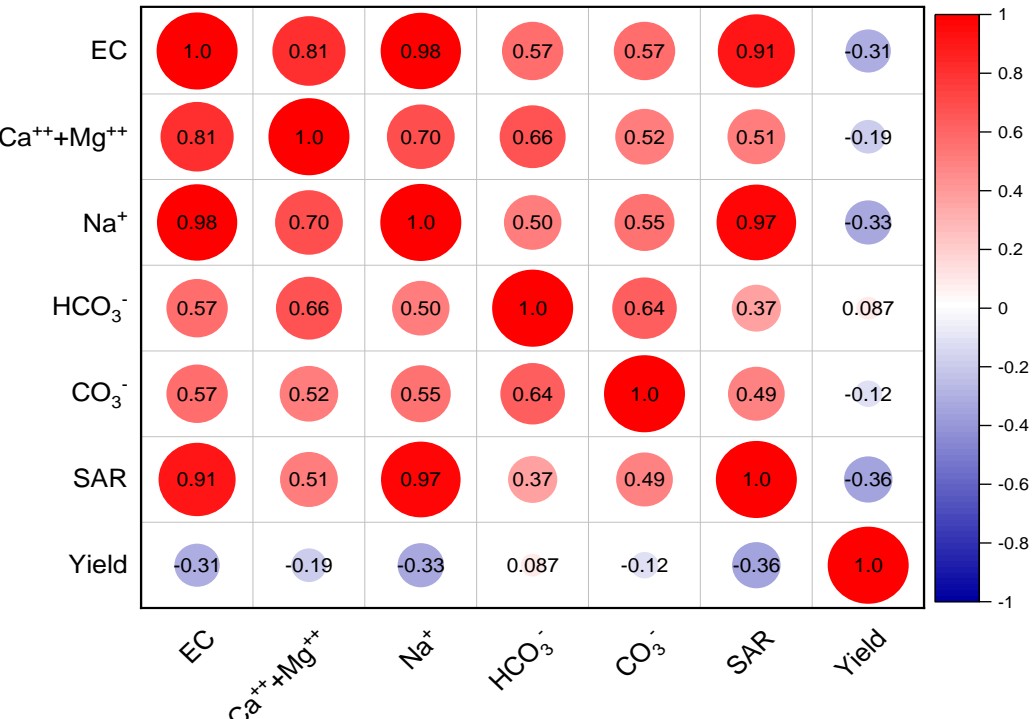

**Figure 8.** Correlation analysis among irrigation water parameters and cotton yield.

## 4. Discussion

Irrigation of crops with brackish water is directly linked to increasing salt accumulation in the rhizosphere and loss of crop productivity [29]. The continuous use of saline water affects the physico-chemical attributes of soil health with a subsequent decline in crop yield [30,31]. In the current study, we have found a large variation in the quality of irrigation water collected from Tehsil Fort Abbas. The change in EC, SAR, and RSC values of collected irrigation water from all sites indicates the level of suitability and associated limitations on the yield of the cotton crop.

### 4.1. Groundwater Quality in Relation to EC, SAR, RSC

The study showed that the tube well water of the large area was unfit due to the higher EC, SAR, and RSC. The EC values of samples taken from the selected locations indicated that water has almost the same ionic composition in the studied areas. The EC value was recorded in the range of 615 to10,490 meq $L^{-1}$. These results are also in agreement with the earlier study conducted by Riaz et al. [4], in which the EC values varied from 0.031 to 15.39 meq $L^{-1}$. The small but non-significant variation in EC of irrigation groundwater is affected by the composition of all the constituents available in ionic forms, notably the hydrogen and hydroxyl ions [32]. Non-significant variation in tested samples was attributed to either better quality groundwater or due to the source's proximity to the water channel [33]. Crop development and yield may be limited by changes in the water's chemical attributes collected from each site. Long-term irrigation of such water may produce soil salinity and deterioration, which can have a negative effect on agricultural

yields, according to previous research described by Hammam and Mohamed [34] and Singh [35].

All the water samples showed a wide range of SAR values (0.65–24.62 mmol/L). Significant variation was noticed in the sodium absorption ratio of irrigation groundwater samples. Some sampling locations in the study area showed SAR values above the permissible limits as described by the Food and Agriculture Organization (FAO). Soil EC and SAR increase due to continuous irrigation with brackish water, which eventually degrades soil's physical and chemical properties [36]. The increase in salinity and SAR values is the primary cause of reduced crop growth and production in brackish water [37,38]. The variation in the SAR of irrigation groundwater samples was expected to be the lowest if proper management practices were employed [39]. Statistically, the highest RSC value (5.44 meq $L^{-1}$) while the minimum RSC value (0.02 meq $L^{-1}$) was found in water samples collected in sampling locations.

High amounts of limestone and magnesium carbonate may also be attributed to a higher level of RSC in these areas [40]. The problem of RSC becomes more serious due to the presence of mineral material rich in certain elements [41]. Another possible source of increased RSC is the presence of inorganic materials in the groundwater [42]. A previous study conducted by Riaz et al. [4] confirmed RSC values up to 43.3 meq $L^{-1}$ in samples of groundwater from the Bahawalpur region. Similarly, significant variation in the chloride contents was noticed in the irrigation groundwater samples from all of the sites. The increase in parameters such as RSC and $Ca^{2+}$ is linked to other soil issues that arise from irrigation with brackish water [43]. Because of the lower quality of the irrigation water, soils with high RSC [4] show increased sodium concentration. Excessive ion concentrations damage soil structure and characteristics in addition to harming plants [44]. Most of the water samples showed maximum chloride contents, and some were above the permissible limits as described by the FAO safe irrigation groundwater. The main reason behind this might be due to the geogenic sources and the chemical weathering of carbonate materials in the local aquifer [45,46].

### 4.2. Groundwater Quality and Heavy Metals

The current study showed a variation of six types of heavy metals (Cu, Zn, Fe, Mn, Cd, and Pb) that were measured in the irrigation groundwater samples collected from different locations in the Fort Abbas area. Similar findings were also observed by Charvalas et al. [47], who assessed heavy metals in the groundwater of Greece and reported the levels of the most frequently detected elements, including Cu, Zn, Fe, Mn, Cd, and Pb. Recently, Kubier et al. [43] reported the heavy metal residues in irrigation groundwater resources to be Fe, Mn, Cd, and Pb. The heavy metal levels in irrigation groundwater also depend upon their chemical properties and types of source minerals, which affect solubility and absorption of pesticides resulting in pesticide accumulation, migration, and diffusion. It can be seen that water resources are currently suffering from widespread pesticide pollution [48].

### 4.3. Groundwater Quality and Cotton Yield

Findings of the current study revealed that the lint yield of cotton is affected by the presence of pollutants in the groundwater. The observed decrease in cotton lint production attributable to water quality is positively linked with water characteristics [49]. In our investigation, using brackish water reduced the amount of lint produced. High $Na^+$, osmotic shifts in the soil, and limited mineral nutrient uptake from the soil can all have a substantial negative impact on cotton growth and output [50]. Additionally, the quantity and frequency of irrigation were major factors in salt buildup in the soil, and a year of continuous irrigation with brackish water led to a significant rise in salt concentration [51]. Our results depicted that use of such irrigation water having higher levels of EC, SAR, and RSC decreased cotton lint yield. Earlier studies reported that irrigation with saline water decreased water uptake and $CO_2$ assimilation. Due to the effect of these processes, plant productivity is affected [52,53].

## 5. Conclusions

Monitoring of water quality has emerged as one of the most important concerns in irrigated agriculture. In the study area, high sodium ion and bicarbonate ion concentrations are affecting groundwater quality and causing environmental risks. It was obvious that 84% of samples were unfit due to the high EC, 26% samples were due to SAR, 6% samples were due to RSC, 88% samples were due to Cu, 18% were due to Zn, 13% were due to Fe, 98% were due to Mn, 99.7% were due to Cd, and 95% were unfit due to Pb concentration. The SAR and RSC values indicate that the water quality for irrigation is not fit, and with the passage of time, water quality is degrading, possibly due to intensive fertilizer application and other anthropogenic activities. Furthermore, there was an inverse relationship between groundwater quality and boring depth. It was evident from these results that the use of sub-standard groundwater could reduce cotton crop production and the income of farmers if used without proper precautions. The prolonged use of such water for irrigation purposes may pose soil salinity and sodicity problems. Therefore, it is recommended that the environmental protection agencies and the irrigation department should develop policies for groundwater quality protection and that areas having higher SAR and RSC values should use groundwater mixed with the canal water to reduce the hardness. Gypsum may be used as a softener for irrigation. Efficient irrigation technologies should be provided to the farmers so that they can use available water efficiently.

**Author Contributions:** Conceptualization, Q.u.Z. and U.R.; methodology, M.R.H. and K.A.; software, U.R., K.S. and A.A.A.K.; formal analysis, M.R.H. and U.R.; investigation, Q.u.Z.; resources, U.R.; data curation, M.R.H.; writing—original draft preparation, M.R.H., B.A., H.A. and K.H.A.; writing—review and editing, Q.u.Z., U.R., B.A. and A.T.A.; supervision, Q.u.Z. and U.R.; project administration, U.R. All authors have read and agreed to the published version of the manuscript.

**Funding:** This research received no external funding.

**Institutional Review Board Statement:** Not applicable.

**Informed Consent Statement:** Not applicable.

**Data Availability Statement:** Not applicable.

**Acknowledgments:** We are thankful to Umair Riaz, Soil and Water Testing Laboratory for Research Bahawalpur, Punjab Government, Pakistan, for providing the research facilities.

**Conflicts of Interest:** The authors declare no conflict of interest.

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
