# Peer review of "Consortium between Groundwater Quality and Lint Yield in Cotton Belt Areas"

_water, doi:10.3390/w14193136_

Round 1
Reviewer 1 Report
In introduction section, please emphasize significance of the study. The objective also needs to be clear, so rewrite the objective. Please clearly state the innovations and significance of the study.
I recommend delete table 1.
Change green to white of the background of Figure 2-4
The Fig 7 was blurry.
It is noted that your manuscript needs careful editing by someone with expertise in technical English editing paying particular attention to English grammar, spelling, and sentence structure so that the goals and results of the study are clear to the reader.
The subtitle of discussion should be more meaningful.
The conclusion repeats the content of results and discussion. It is essential to rewrite the conclusion to summary primary results, primary viewpoint, shortcomings, academic contribution, and some suggestion.
Author Response
Dear Reviewer,
Thank you so much for giving us an opportunity to revise the manuscript, Hope so the changes suggested by you, will be helpful in improving the quality and readability of manuscript. We have tried our best to add the suggestions from your side in the highlighted form.
If you have any query, please feel let me know
Qamar uz Zaman

Reviewer 2 Report
This paper lacks novelty, manuscript is is poorly written, and needs structural revisions
The results is limited to no (%) samples that are fit-unfit, with no merit of discussion is detected in text
Author Response

(The authors gave the same response as above.)

Reviewer 3 Report
In this research, The authors are studying the consortium between ground water quality and lint yield in cotton belt areas of southern punjab. this study was conducted to explore the water quality status of Tehsil Fort Abbas along with cotton yield. However, there were still some shortcomings:
1 In figure 5, it is recommended to add standard error data
2 In Table 3, it is suggested to add the relevant data of cotton irrigation water quality in each district.
3 There is no substantive discussion in the first paragraph of the discussion, and it is suggested that it should be integrated with the first paragraph of the introduction.
4 Lack of data support in the conclusion part.
5 The references of this paper suggest to quote the latest research results in the last 3 years, especially in 2022.
Author Response

(The authors gave the same response as above.)

Round 2
Reviewer 1 Report
The authors have properly revised their manuscript in response to my previous comments. Thus, I would like to make my suggestion on accepting it for publication.
Author Response
Thank you very much for giving us a chance to respond to the reviewers’ comments for 2nd time. We have made every effort to address each reviewer’s concerns as clearly and succinctly as possible, as demonstrated in the following pages. Thank you once again for this opportunity and please let us know if you have any questions or concerns.

Reviewer 2 Report
To bring this manuscript to acceptable form in Water, I advise the authors to do the following:
Abstract
1- EC is given in meq/L (Is this true??). For example 500 meq/L of Na is equivalent to 18 dS/m (this is too high). EC measures total dissolved ions in water and measured in mmohs/cm or dS/m
Please revise all related EC values (figs and tables)
Use million cubic meters -MCM after first abbreviation and through text
Compare your values to permissible limits by (e.g. FAO, Ayers and Westcot, 1984) for lint production
No need to mention which chak no. are exceeding the limits, Just mention as % of the total samples
Materials and methods:
SAR is in meq/L NOT mmole/L (to use mmoles you need to delete 2 in equation)
Results
use descriptive stat for all parameters (in new table: avg, sd, cv, skew, Q1-Q3)
then discuss % of samples that are fit-unfit
Use this for metals as well (no need to have subtitles for 5 metals, this can be discussed under one section (metals)
All related discussions can be shorten after
Stress more on correlation of crop yield and other parameters.
A relation between EC (correct for unit) and yield could be established
Fig 7 can be improved (Please see unit for EC and others (correct)
ALL related discussion should be based on the criteria for irrigation water which should be mentioned correctly in related table
Chak no should be deleted from your discussion and replaced with areas found in your map
Author Response

(The authors gave the same response as above.)
